# Transport Properties in Magnetized Compact Stars

**Toshitaka Tatsumi [1],\* and Hiroaki Abuki [2]**

[1] Institute of Education, Osaka Sangyo University, 3-1-1 Nakagaito, Daito, Osaka 574-8530, Japan

[2] Department of Physics, Aichi University of Education, Hirosawa 1, Kariya 448-8542, Japan; abuki@auecc.aichi-edu.ac.jp

\* Correspondence: tatsumitoshitaka@gmail.com

**Abstract:** Transport properties of dense quark matter are discussed in the strong magnetic field, $B$. $B$ dependence as well as density dependence of the Hall conductivity is discussed, based on the microscopic Kubo formula. We took into account the possibility of the inhomogeneous chiral phase at moderate densities, where anomalous Hall effect is intrinsic and resembles the one in Weyl semimetals in condensed matter physics. Some theoretical aspects inherent in anomalous Hall effect are also discussed.

**Keywords:** quark matter; chiral symmetry; Hall conductivity; spectral asymmetry; Dirac material





## 1. Introduction

Observations of compact stars have provided us with information about highly dense matter in their cores since their first discovery in 1967. Nowadays, many papers have appeared about the possible presence of quarks and its implications on observations of neutron-star mergers, high-mass stars about $2M_\odot$ or magnetars [1,2]. Magnetars bear a huge magnetic field of $O(10^{15}\mathrm{G})$ in their surface and exhibit unique thermal evolution [3], while the origin of the magnetic field and the surface temperature have not been fully understood yet. Their persistent surface temperature is very high, compared to ordinary pulsars at the same age, and resides well above the standard cooling curve [4]. In order to resolve the issue, we must take into account the thermal conduction as well as the heating mechanism in the presence of a strong magnetic field. One of the authors (T.T.) has suggested a possibility of spontaneous magnetization of quark matter inside cores as a microscopic origin of huge magnetic fields, based on the energetic scale of QCD [5]. Herein, we examine the transport properties of quark matter under a large magnetic field, which provides, we expect, a first step to understanding thermal evolution of magnetars with quark core.

Because the transport properties are important for thermal evolution of ordinary pulsars, there have been many works [6]. However, the microscopic treatment of thermal conductivity may include many subtle points, such as relativistic effects, quantum mechanical effects (including the Shubnikov–De Haas effect due to discretized Landau levels) in the magnetic field. Herein, we re-examine these points for relativistic fermions, such as quarks or electrons. For example, electrons become relativistic in the inner crust of neutron stars, and we must use the Dirac equation to describe them. It should be interesting to see such Dirac electrons become important in modern condensed matter physics [7,8], where some topological materials have been also discussed. If such topological materials develop in the inner crust, we must carefully discuss the transport properties of electrons there.

The basic framework is the same for conducting electrons and quarks to discuss the transport phenomena; we briefly summarize it for electrons for simplicity. For the electric

current $\mathbf{j}$ and the energy current $\mathbf{j}_E$, phenomenological-transport equations should read for charge-carrying electrons:

$$\mathbf{j} = L_{11}\left[\mathbf{E} - \frac{T}{e}\nabla\left(\frac{\mu}{T}\right)\right] + L_{12}\left[T\nabla\left(\frac{1}{T}\right) - \nabla\phi_g\right], \tag{1}$$

$$\mathbf{j}_E = L_{21}\left[\mathbf{E} - \frac{T}{e}\nabla\left(\frac{\mu}{T}\right)\right] + L_{22}\left[T\nabla\left(\frac{1}{T}\right) - \nabla\phi_g\right], \tag{2}$$

with the electric field, $\mathbf{E} = -\nabla\phi$. $\phi_g$ is the fictitious "gravitational" potential introduced by Luttinger [9], with which one can apply the linear-response theory for thermal conductivity. The conductivity tensor $\sigma$ and the thermal conductivity $\kappa$ are constructed by combining the matrix elements $L_{ab}$ (second rank tensors) as

$$\sigma = L_{11,} \tag{3}$$

$$\kappa = T^{-1}\left(L_{22} - L_{21}^{-1}L_{11}L_{12}^{-1}\right). \tag{4}$$

The matrices $L_{ab}$ are related to each other, and we confirm that the Wiedemann–Franz law holds at low temperature,

$$\kappa = LT\sigma, \tag{5}$$

with the Lorentz number, $L = \frac{1}{3}(\pi k_B/e)^2$. Thus, we need information about the conductivity $\sigma$ at vanishing temperature ($T = 0$) to obtain thermal conductivity $\kappa$ at low temperature. In general, $\sigma$ or $\kappa$ has off-diagonal components, corresponding to the Hall effect.

The evaluation of the transport coefficients $L_{ab}$ can be done by the Boltzmann equation or the Kubo formula in a microscopic way. Here, we use the Kubo formula within the linear-response theory [10]. It is to be noted that quark matter contains flavor and color degrees of freedom, and we must carefully sum up their contributions.

For quarks, we see an interesting possibility of the anomalous Hall effect (AHE) [11,12] at moderately high densities. Recently possible appearance of the inhomogeneous chiral phase (iCP) has been suggested to show up in some region of the QCD phase diagram in the temperature ($T$)-baryon-number chemical potential ($\mu$) plane (see Figure 1). Therefore, if quark matter is realized in the core region of compact stars, there may be iCP developed. The physical properties of the iCP phase in various situations, including those in the presence of magnetic field, have been extensively studied [13–18]. The dual chiral density wave (DCDW) is one type of iCP, which is a kind of density wave and specified by the scalar and pseudoscalar condensates with spatial modulation [13],

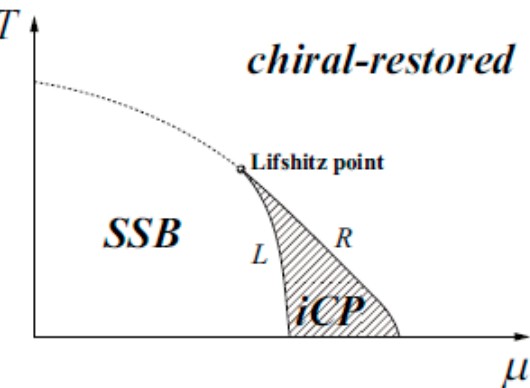

**Figure 1.** Schematic picture of the QCD phase diagram in the $(T, \mu)$-plane. Inhomogeneous chiral phase (iCP) may appear near the chiral transition, and spontaneously symmetry broken (SSB) phase, chiral-restored phase, and iCP meet at the triple point called the Lifshitz point.

$$\langle \overline{\psi}\psi \rangle = \Delta\cos(\mathbf{q}\cdot\mathbf{r}), \tag{6}$$

$$\langle \overline{\psi} i \gamma_5 \tau_3 \psi \rangle = \Delta \sin(\mathbf{q} \cdot \mathbf{r}). \tag{7}$$

The order parameters are the amplitude $\Delta$ and the wave vector $\mathbf{q}$. We shall see the DCDW phase shares similar physical properties with Weyl semimetals (WSM) in condensed-matter physics [19].

## 2. Brief Review of AHE in the DCDW Phase

### 2.1. Dual Chiral Density Wave

AHE and its implication to axion electrodynamics have been discussed with a focus on its relation to the DCDW phase [20,21]. For flavor symmetric $u, d$ quark matter, the single-particle energy of quarks can be easily obtained in the DCDW phase, when the Nambu–Jona-Lasinio (NJL) model is used as an effective model of QCD at low-energy scales,

$$L_{\mathrm{NJL}} = \overline{\psi} \left( i \gamma^\mu \partial_\mu - m \right) \psi + G[(\overline{\psi}\psi)^2 + (\overline{\psi} i \gamma_5 \boldsymbol{\tau} \psi)^2]. \tag{8}$$

Using the ansatz (1) for the chiral condensates, the effective Lagrangian reads

$$L_{MF} = \overline{\psi} \left( i \gamma^\mu \partial_\mu - \frac{1 + \gamma_5 \tau_3}{2} M(z) - \frac{1 - \gamma_5 \tau_3}{2} M^*(z) \right) \psi - \frac{|M(z)|^2}{4G}, \tag{9}$$

$$M(z) = -2G\Delta e^{iqz} \left( \equiv M e^{iqz} \right), \tag{10}$$

in the chiral limit, $m_c = 0$, under the mean-field approximation [13]. One may rewrite it in a simple form,

$$L_{MF} = \overline{\psi}_W \left( i \gamma^\mu \partial_\mu - M - \frac{1}{2} \gamma_5 \tau_3 \gamma_\mu q^\mu \right) \psi_W - G\Delta^2 \tag{11}$$

with $M = -2G\Delta$ and the space-like vector $q^\mu = (0, \mathbf{q})$ by the use of the Weinberg transformation, $\psi_W \equiv \exp[i\gamma_5\tau_3/2x(\mathbf{r})]\psi = \exp[i\gamma_5\tau_3\mathbf{q}\cdot\mathbf{r}/2]\psi$. We can see that the amplitude of DCDW provides the dynamical mass for the newly defined quarks (quasiparticles) described by $\psi_W$, while the wave vector induces the axial-vector mean-field applied to them.

The single-particle energy can be easily extracted,

$$E_{\epsilon=\pm 1,\, s=\pm 1}(p) = \epsilon \sqrt{E_{\mathbf{p}}^2 + \frac{\mathbf{q}^2}{4} + s\sqrt{(\mathbf{p}\cdot\mathbf{q})^2 + M^2\mathbf{q}^2}} \tag{12}$$

with $E_{\mathbf{p}} = \sqrt{\mathbf{p}^2 + M^2}$ for each flavor, where $\epsilon$ and $s$ denote the particle–antiparticle and spin degrees of freedom, respectively. Accordingly, this form suggests anisotropy of the Fermi sea in the momentum space: it deforms about the direction of $\mathbf{q}$ in the axial-symmetric manner.

In Figure 2, we plotted the energy surface in the momentum space. One can see that there is a gap between negative and positive energies in the case with $q/2 < M$ (Figure 2a), while in the opposite case with $q/2 > M$, there are two Weyl points $\mathbf{p} = (0, 0, \pm K_0)$, with $K_0 = \sqrt{(q/2)^2 - M^2}$ where the gap vanishes, as seen in Figure 2b. The thermodynamic potential can be derived by the single-particle energies, and the parameters $(q, M)$ are determined by the minimization of the thermodynamic potential: it has been shown that the relation $q/2 > M$ holds in the DCDW phase [13].

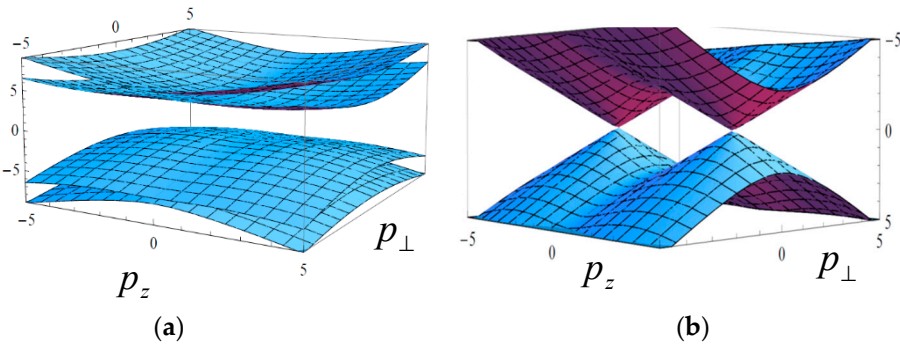

**Figure 2.** Energy surfaces for different regions of the parameters: Left panel (**a**) is for the case $M > q/2$, while right panel (**b**) is for the case $M < q/2$.

Here some resemblance with WSM is worth mentioning [19]. The effective Lagrangian (3) is of the same form as in WSM, by replacing $q/2$ by the strength of spin-splitting $b$ and the dynamical mass $M$ by the spin–orbit coupling strength $m$. The relation, $b > m$ holds in WSM. The important difference is that the values of $(q, M)$ are dynamically determined as a consequence of chiral symmetry breaking in the DCDW phase, while the parameters $(b, m)$ can be controlled with the experimental setup for WSM. Positive-energy states correspond to the conduction band, while the negative-energy states correspond to the valence band. Because some positive energy states are filled in the DCDW phase, one might call it a "Weyl metallic state" [22]. Therefore, we discuss, hereafter, the transport properties of dense quark matter, referring to WSM as a guiding principle.

### 2.2. Anomalous Hall Effect

One of the interesting transport properties in WSM may be the anomalous Hall effect (AHE) [19]: the Hall current flows even in the absence of magnetic field in response to the external electric field. Accordingly, we shall see AHE in the DCDW phase. We can derive the anomalous Hall conductivity by way of the Kubo formula, considering a linear response to a tiny electric field [20]. It is given by the integral of the Berry curvature in the momentum space $\mathbf{b}(\mathbf{p})$,

$$\sigma_{xy} = e^2 \int \frac{d^3 p}{(2\pi)^3} b_z(\mathbf{p}) f(E_\mathbf{p}) \tag{13}$$

with $f(E_p)$ being the Fermi–Dirac distribution function. The Berry curvature is defined in terms of the eigenfunctions $u_\mathbf{p}$ as

$$\mathbf{b}(\mathbf{p}) = -i \nabla_\mathbf{p} \times \langle u_\mathbf{p} | \nabla_\mathbf{p} | u_\mathbf{p} \rangle \tag{14}$$

and it reads

$$b_{s,z}(\mathbf{p}) = \frac{-1}{2E_{\epsilon=+1,s}^3} \left( sE_0 + \frac{q}{2} \right) \tag{15}$$

with $E_0 = \sqrt{p_z^2 + M^2}$. It is interesting to see that the Berry curvature looks like the magnetic field from the Dirac monopole located at each Weyl point in the momentum space. The contribution from the negative-energy sea apparently diverges and needs a relevant regularization [23]. We see that it is appropriate to use the proper-time method or the heat-kernel method. Applying the proper-time regularization, we find

$$
\begin{aligned}
\sigma_{xy}^{\text{Dirac}} &= \lim_{\Lambda \to \infty} \frac{e^2}{2\Gamma\left(\frac{3}{2}\right)} \sum_{s=\pm 1} \int \frac{d^3 p}{(2\pi)^3} \left( sE_0 + \frac{q}{2} \right) \int_{\Lambda^{-2}}^{\infty} d\tau \tau^{-\frac{1}{2}} e^{-\tau\left(sE_0 + \frac{q}{2}\right)^2} \\
&= \frac{e^2}{4\pi^2} \sum_{s=\pm 1} \int_0^{\infty} dp_z \operatorname{sign}\left( sE_0 + \frac{q}{2} \right) \\
&\quad - \frac{e^2}{4\pi^{\frac{5}{2}}} \lim_{\Lambda \to \infty} \sum_{s=\pm 1} \int_0^{\infty} dp_z \operatorname{sign}\left( sE_0 + \frac{q}{2} \right) \int_0^{\Lambda^{-2}} d\tau \tau^{-\frac{1}{2}} e^{-\tau\left(sE_0 + \frac{q}{2}\right)^2}.
\end{aligned} \tag{16}
$$

The second term is evaluated to give $-e^2 q/(2\pi)^2$, while the first term gives different values, depending on the values of the wave vector: for $q/2 < M$ it vanishes, while it gives

$$\frac{e^2}{2\pi} \times \int_{-K_0}^{K_0} \frac{dp_z}{2\pi} = \frac{e^2 K_0}{2\pi^2}, \tag{17}$$

for $q/2 > M$. It is interesting to note that the quantity $e^2/2\pi$ is a topological number and nothing else but the Hall conductivity for 2D quantum Hall systems with the Chern number $\nu = 1$. Therefore, we may regard the DCDW phase as a stack of 2D quantum Hall systems to the 3rd direction [24]. The anomalous Hall conductivity reads

$$\sigma_{xy}^{\text{Dirac}} = \frac{e^2}{4\pi^2}\left(-q + 2K_0\theta\left(K_0^2\right)\right), \tag{18}$$

where $K_0^2 \equiv \left(\frac{q}{2}\right)^2 - M^2$. We can immediately see that $\sigma_{xy}^{\text{Dirac}} \to 0$ as $M \to 0$, because in this limit $K_0 \to q/2$. It is reasonable for $\sigma_{xy}$ to be vanishing for $M = 0$, implying no AHE for the normal phase. We shall see that the origin of the first term comes from axial anomaly. On the other hand, the anomalous Hall conductivity in WSM is given only by the second term without the anomaly term: $\sigma_{xy}^{\text{WSM}} = e^2 K_0/\left(2\pi^2\right)$ for $b > m$, while it is vanishing for $b < m$. In fact, Goswami and Tewari have shown the bulk-boundary correspondence by explicitly constructing the surface states: the surface states exist only for $b > m$, and the system is an insulator for $b < m$ [25].

### 2.3. Fermi Sea Contribution

Let us now have a quick look at the Fermi sea contributions. For cases (a) $\mu < \frac{q}{2} - M$, (b) $\frac{q}{2} - M < \mu < \frac{q}{2} + M$, (c) $\frac{q}{2} + M < \mu$, we find [21]

$$\left(\sigma_{xy}^{\text{Fermi}}\right)_{\text{DCDW}} = \frac{e^2}{(2\pi)^2}\begin{cases} \frac{\left(\mu+\frac{q}{2}\right)^2}{2\mu}\sin\theta_+ - \frac{\left(\mu-\frac{q}{2}\right)^2}{2\mu}\sin\theta_- - \frac{M^2}{4\mu}\ln\frac{(1+\sin\theta_+)(1-\sin\theta_-)}{(1-\sin\theta_+)(1+\sin\theta_-)} - 2K_0, & (a) \text{ or } (c) \\ \frac{\left(\mu+\frac{q}{2}\right)^2}{2\mu}\sin\theta_+ - \frac{M^2}{4\mu}\ln\frac{(1+\sin\theta_+)}{(1-\sin\theta_+)} - 2K_0, & (b) \end{cases} \tag{19}$$

where $\sin\theta_\pm = \sqrt{1 - M^2/\left(\mu \pm \frac{q}{2}\right)^2}$. $\sigma_{xy}^{\text{Fermi}}$ measured in the unit of $\sigma_{xy}^{\text{Dirac}}$ is plotted as a function of $\mu$ in Figure 3. It is interesting to note that $\left(\sigma_{xy}^{\text{Fermi}}\right)_{\text{DCDW}} \to 0$ as $M \to 0$ for an arbitrary density; this means that there is no AHE in the chiral restored phase, irrespective of the value of wave number $q$.

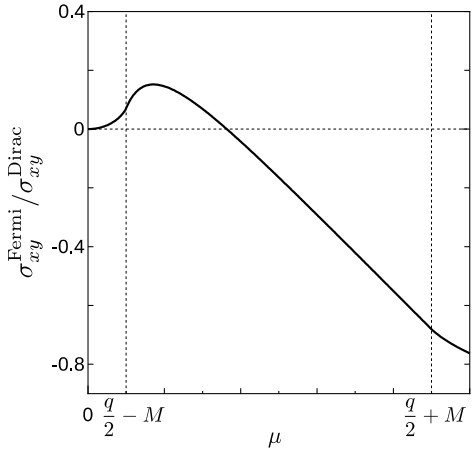

**Figure 3.** Density dependence of $\left(\sigma_{xy}^{\text{Fermi}}\right)_{\text{DCDW}}$.

## 3. Hall Conductivity in the Presence of Magnetic Field

### 3.1. Anomalous Hall Conductivity

Let us now discuss how the situation changes when the system is immersed in an external magnetic field [26]. Using Equation (3), the effective Hamiltonian for quasiparticles in the DCDW phase is given as $\psi_W^\dagger H_{MF} \psi_W + G\Delta^2$, where

$$H_{MF} = \boldsymbol{\alpha} \cdot (\mathbf{p} - Q\mathbf{A}) + \beta M + \mathbf{q} \cdot \boldsymbol{\alpha} \, \gamma_5 \tau_3 / 2 \tag{20}$$

with $Q = \text{diag}(e_u, e_d)$ ($e_u = 2e/3$, $e_d = -e/3$). Because it has been shown that $\mathbf{q} \parallel \mathbf{B}$ is the most favorable configuration [27], we set magnetic field along the $z$-axis without loss of generality.

The effective Hamiltonian can be easily diagonalized, and resulting eigenvalues are

$$E_{n,s,\epsilon}^f(p_z) = \epsilon \sqrt{\left(s\sqrt{M^2 + p_z^2} - \frac{q}{2}\right)^2 + 2\left|e_f\right| Bn}, \; n = 1, 2, 3, \dots,$$

$$E_{n=0,\epsilon}(p_z) = \epsilon \sqrt{M^2 + p_z^2} - \frac{q}{2}, \tag{21}$$

for each flavor $f$, where $\varepsilon = \pm 1$ denotes the particle–antiparticle states, and $s = \pm 1$ specifies the spin degree of freedom. Note that there is no spin degree of freedom for the lowest Landau level (LLL): dimensional reduction occurs for LLL, and the eigenspinor is represented by two components. The energy spectra are depicted in Figure 4 for two cases. Note that, the external magnetic field makes the DCDW phase extend to lower densities, and both cases are realized in the presence of the magnetic field [27,28], in contrast to the previous situation discussed in Section 3.

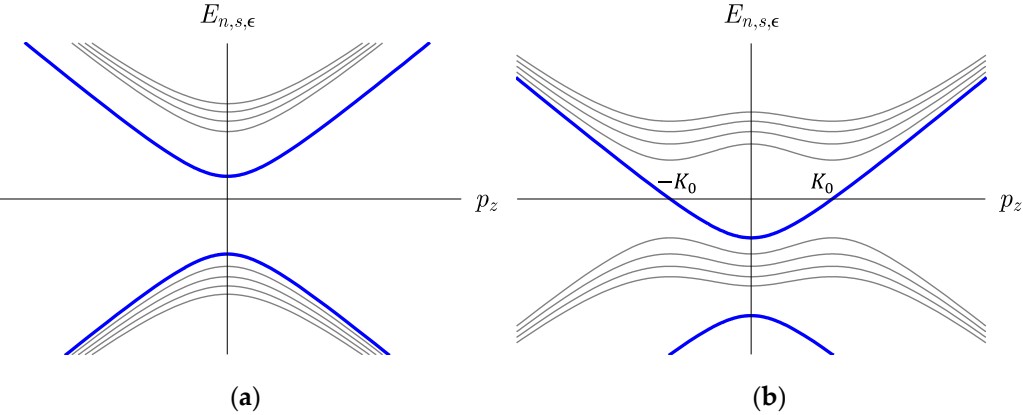

|     |     |
| --- | --- |
| **(a)** | **(b)** |

**Figure 4.** Energy spectra in the presence of magnetic field. Left panel (**a**) is for the case of $M > \frac{q}{2}$, while right panel (**b**) is for the case of $M < \frac{q}{2}$. Lowest Landau level (LLL) for each case is denoted by the bold lines.

The conductivity tensor $\sigma$ consists of two kinds of the matrix elements, the diagonal ones $\sigma_{xx} = \sigma_{yy}$, $\sigma_{zz}$ and the off-diagonal one $\sigma_{xy}$. The latter one implies the Hall effect and becomes important for small impurities in the presence of magnetic field, compared to the longitudinal conductivity $\sigma_{xx} = \sigma_{yy}$. Next, we consider the Hall conductivity.

The Kubo formula gives the Hall conductivity $\sigma_{xy}$ at $T = 0$ in the presence of a magnetic field [29]. Streda has further divided $\sigma_{xy}$ into the following two terms [30],

$$\sigma_{xy} = \sigma_{xy}^{\text{I}} + \sigma_{xy}^{\text{II}}, \tag{22}$$

where the first term $\sigma_{xy}^{\text{I}}$ vanishes when the chemical potential is located between the energy gap where the density of states is vanishing. This situation occurs for topological insulators, and also in the nearly clean 2D quantum Hall system in a strong magnetic field. On the other hand, in the case of finite density of states at the Fermi energy, the dissipative effects are no longer to be ignored. In this case, $\sigma_{xy}^{\text{I}}$ depends on the details of matter. In

fact, we can see the classical Drude–Zener relation $\sigma_{xy}^{\text{I}} = -\omega_c \tau \sigma_{xx}$, by using arbitrary energy-dependent self-energy [30], where $\omega_c$ is the cyclotron frequency and $\tau$ the life-time. Note that $\sigma_{xy}^{\text{I}}$ remains finite even in the dissipation-less limit, $\tau \to \infty$. On the other hand, the second non-dissipative term has no classical analogy, which can be expressed by the use of the number of states under the energy $E$, $N^f(E)$, for each flavor to represent the quantum effect,

$$\sigma_{xy}^{\text{II}} = -\sigma_{yx}^{\text{II}} = \sum_f e_f \left.\frac{\partial N^f}{\partial B}\right|_{E=\mu}. \tag{23}$$

It may be worth noting that for quantized Hall effect in 2D Hall systems, $\sigma_{xy}^{\text{I}} = 0$, $\sigma_{xy}^{\text{II}} = -e^2 N/(2\pi)$ with $N$ being the integer called the Landau-level filling factor, that is, the number of Landau levels below the Fermi energy [30]. Noting $N = (n_e L_x L_y)/d$ with $n_e$ being the electron density and $d = \frac{eB}{2\pi\hbar} L_x L_y$ being the single Landau-level degeneracy factor, we see the formula reduces to $\sigma_{xy}^{\text{II}} = -en_e/B$. Because $n_e = \frac{eB}{2\pi\hbar} N$, the formula $\sigma_{xy}^{\text{II}} = -e\frac{\partial n_e}{\partial B}$ is actually satisfied. On the other hand, in 3D Dirac materials with $b = 0$, it has been shown that $\sigma_{xy} = -en_e/B$ [7], provided that the effect of impurities can be neglected. We do not discuss $\sigma_{xy}^{\text{I}}$ here and mainly focus on the quantum contribution $\sigma_{xy}^{\text{II}}$, where we should see some topological effect.

Writing the baryon number operator in the normal ordering form, $\hat{N}_f = \frac{1}{2} \int d^3x [\psi_f^\dagger(x), \psi_f(x)]$ ($f = u$ or $d$), we find $N(E) = N_{\text{norm}}(E) + N_{\text{anom}}$ by counting the number of the eigenstates below energy $E$. The quasiparticle density of states can be written as

$$D_{\text{DCDW}}^f(\lambda) = N_c \frac{|e_f| B}{(2\pi)^2} \sum_{\epsilon=\pm1} \int_{-\infty}^{\infty} dp_z \left( \delta(\lambda - E_{n=0,\epsilon}(p_z)) + \sum_{s=\pm1} \sum_{n=1}^{\infty} \delta(\lambda - E_{n,s,\epsilon}^f(p_z)) \right), \tag{24}$$

in the DCDW phase, and thereby $N_{\text{norm}}(E)$ reads

$$N_{\text{norm}}(E) = \sum_f N_{\text{norm}}^f(E) = \sum_f \int_0^E D_{\text{DCDW}}^f(\lambda) d\lambda. \tag{25}$$

Therefore, the contribution to the conductivity from the Fermi sea can be written as

$$\sigma_{xy}^{\text{Fermi}} = \sum_f e_f \left.\frac{\partial N_{\text{norm}}^f(E)}{\partial B}\right|_{E=\mu}. \tag{26}$$

The second one is the anomalous quark number and can be written as

$$N_{\text{anom}} = -\frac{1}{2} \int_{-\infty}^{\infty} \text{sign}(\lambda) \text{Tr}\delta(\lambda - H) d\lambda. \tag{27}$$

We can see that it originates from the spectral asymmetry and is closely related to a topological quantity, the $\eta$ invariant introduced by Atiyah–Patodi-Singer [31], $N_{\text{anom}} = -\frac{1}{2} \sum_f \eta_H^f$. The expression (18) is not well defined as it is and must be properly regularized to extract the physical result. Using the gauge-invariant regularization, we can evaluate the $\eta$ invariant [32],

$$\eta_H^f = N_c \frac{|e_f| B q}{2\pi^2} - \theta\left(\frac{q}{2} > M\right) \frac{|e_f| B}{\pi^2} N_c \sqrt{\left(\frac{q}{2}\right)^2 - M^2}. \tag{28}$$

We can see from the first term that the $\eta$ invariant is related to axial anomaly. The second term correctly cancels the anomalous contribution from the first term in the limit $M \to 0$.

### 3.2. Axial Anomaly

Introducing chemical potential $\mu$ as a fictitious gauge field $B^\mu = (\mu, 0)$ coupling with quark number, the mean-field Lagrangian should be generalized from Equation (2) to be

$$L_{MF} = \bar{\psi}\left(i\gamma^\mu D_\mu - \frac{1+\gamma_5\tau_3}{2}M(z) - \frac{1-\gamma_5\tau_3}{2}M^*(z)\right)\psi - \frac{|M(z)|^2}{4G}, \tag{29}$$

where the covariant derivative is now given by $D^\mu = \partial^\mu - i(QA^\mu + B^\mu)$. Then, we find an anomaly term in the action after the Weinberg transformation [33,34], $\psi_W \equiv \exp[i\gamma_5\tau_3/2x(\mathbf{r})]\psi = \exp[i\gamma_5\tau_3\mathbf{q}\cdot\mathbf{r}/2]\psi$,

$$\begin{aligned} S_{\text{ano}} &= \frac{1}{16\pi^2}\int d^4x\, x(\mathbf{r})(e^2 F\widetilde{F} - 6eG\widetilde{F}) \\ &= \frac{1}{8\pi^2}\int d^4x\, \partial_\mu x(\mathbf{r})(e^2 A_\nu \widetilde{F}^{\mu\nu} - 6eB_\nu\widetilde{F}^{\mu\nu}), \end{aligned} \tag{30}$$

with $G^{\mu\nu} = \partial^\mu B^\nu - \partial^\nu B^\mu$. The first term represents the standard axial anomaly and vanishes in the absence of electric field, while the second term produces

$$\mu n_{\text{anom}} \equiv -3\mu\frac{e\mathbf{B}\cdot\nabla x(\mathbf{r})}{4\pi^2}. \tag{31}$$

We can see an anomalous quark number appear. Such an anomalous quark number becomes the same as the one given by the first term of the $\eta$ invariant (19). Note that this result has been explicitly verified by evaluating the $\eta$ invariant by using the adiabatic expansion for the quark propagator a la Goldstone and Wilczek [35].

Accordingly, the anomalous Hall conductivity can be given as

$$\begin{aligned} \sigma_{xy}^{\text{anom}} &= -\frac{1}{2}\sum_f e_f \frac{\partial\eta_H^f}{\partial B} \\ &= -\frac{e^2 q}{4\pi^2} + \frac{e^2}{2\pi^2}K_0\theta(K_0^2) \\ &= \begin{cases} -\frac{e^2 q}{4\pi^2}, & M > \frac{q}{2}, \\ -\frac{e^2 q}{4\pi^2} + \frac{e^2}{2\pi^2}K_0, & M < \frac{q}{2}, \end{cases} \end{aligned} \tag{32}$$

for $N_c = 3$. Thus, we confirm that $\sigma_{xy}^{\text{anom}}$ coincides with Equation (9). Note that AHE has been also discussed for WSM in the magnetic field [36]: they obtained a different result,

$$\sigma_{xy}^{\text{anom}} = \frac{e^2}{2\pi^2}\sqrt{b^2 - m^2}, \tag{33}$$

in accordance with the bulk-boundary correspondence. The difference between Equations (22) and (23) comes from the regularization. Technically, they did not used a gauge-invariant regularization, while Equation (22) is obtained by the gauge-invariant one.

### 3.3. Fermi Sea Contribution

Let us finally have a look at the Fermi sea contribution $\sigma_{xy}^{\text{Fermi}}$ given in Equation (17). Generally, we must take into account many Landau levels, depending on chemical potential $\mu$ and the strength of the magnetic field $B$ [22]. We can derive an analytic formula in the limit of strong magnetic field. This is performed by restricting the level summation in the density of state to the contribution from the LLL (quantum limit regime). In this approximation we have

$$\begin{aligned} \sigma_{xy}^{\text{II,Fermi,LLL}} &= \frac{e^2}{2\pi^2}\left(p_F\theta(p_F^2) - K_0\theta(K_0^2)\right) \\ &= \begin{cases} \frac{e^2 p_F}{2\pi^2}\theta(p_F^2), & M > \frac{q}{2}, \\ \frac{e^2 p_F}{2\pi^2}\theta(p_F^2) - \frac{e^2}{2\pi^2}K_0, & M < \frac{q}{2}, \end{cases} \end{aligned} \tag{34}$$

where $p_F^2 \equiv \left(\mu + \frac{q}{2}\right)^2 - M^2$. We note that when the system approaches the homogeneous limits $q = 0$, $\sigma_{xy}^{\text{II,Fermi,LLL}} \to \frac{e^2 p_F}{2\pi^2}$. The Fermi momentum $p_F = \sqrt{\mu^2 - M^2}$ is now proportional to the fermion density since the system is dimensionally reduced to 1D. In fact, the fermion density in this limit can be easily evaluated as,

$$n_F = \sum_f n_F^f = N_c \frac{eB}{2\pi^2} p_F, \quad \left(n_F^f = N_c \frac{\left|e_f\right| B}{2\pi^2} p_F\right) \tag{35}$$

where only spin-down u-quarks and spin-up d-quarks in the LLL contribute to the density. Then, we have the standard relation $\sigma_{xy}^{\text{II,Fermi,LLL}} = \frac{e n_F}{3B} \left(= \sum_f \frac{e_f \, n_F^f}{B}\right)$. We can say that in this limit, the $1/B$ scaling of the Hall conductivity totally comes from quantum (non-dissipative) contribution $\sigma_{xy}^{\text{II,Fermi}}$. It should be interesting to see that the dissipative part $\sigma_{xy}^{\text{I,Fermi}}$ has nothing to do with the Hall conductivity in this limit. This result can never be inferred from the classical relation, $\sigma_{xy}^{\text{I}} = -\omega_c \, \tau \sigma_{xx}$, which has been used for the electron conductivity for thermal evolution of neutron stars with strong magnetic fields [6]. Incidentally, the dominance of $\sigma_{xy}^{\text{II,Fermi}}$ over $\sigma_{xy}^{\text{I,Fermi}}$ holds in the limit, $q \to 0$, as in Dirac material: a direct evaluation of $\sigma_{xy}^{\text{I,Fermi}}$ can be easily done to give a null result in the high-field limit.

On the other hand, in the case of weak magnetic field, we may derive the approximated expression by replacing the summation over Landau levels by the continuous integration. In general, we can expand $\sigma_{xy}^{\text{Fermi}}$ as

$$\sigma_{xy}^{\text{II, Fermi}} = \frac{a_{-1}(M, q, \mu)}{B} + a_0(M, q, \mu) + a_1(M, q, \mu)B + O\left(B^2\right). \tag{36}$$

Setting a continuous function $\widetilde{D}(x) = \frac{N_c}{8\pi^2} \int_0^\mu d\lambda \int_{-\infty}^\infty dp_z \sum_{f,\varepsilon,\, s} e_f \delta\left(\lambda - E_{n,s,\varepsilon}^f(p_z)|_{2|e_f|Bn \to x}\right)$, we find the integral expression for $a_{-1}(M, q, \mu)$, which turns out to vanish after the integration by parts:

$$a_{-1}(M, q, \mu) = \int_0^\infty \widetilde{D}(x)dx + \int_0^\infty x\widetilde{D}\prime(x)dx = 0. \tag{37}$$

The first nontrivial term $a_0(M, q, \mu)$ should coincide with $a_0(M, q, \mu) = \left(\sigma_{xy}^{\text{Fermi}}\right)_{\text{DCDW}}$ in Equation (10). We see that Fermi sea contribution to the non-dissipative conductivity vanishes ($\sigma_{xy}^{\text{II, Fermi}} \to 0$) as $B \to 0$. It was shown in [7] for Dirac material ($q \to 0$), the relation $\sigma_{xy} = -en_e/B$ holds, irrespective of the value of $B$. Then, we may conclude that, in the weak field limit, the $1/B$ scaling of the Hall conductivity should come totally from the dissipative part $\sigma_{xy}^{\text{I,Fermi}}$. It is worth mentioning that, such $1/B$ scaling itself is natural when the system has freely moving charge carriers, as is also inferred from the Drude–Zener model [37]. This is a consequence of the fact that active fermions in conduction bands move so that, in the equilibrium state, they do no longer feel the applied electric field $E_y$, which disappears in the comoving frame by the Lorentz transformation. To be specific, let us consider the system with a magnetic field pointing to z-direction $\boldsymbol{B}$ and an electric field pointing to $y$ direction $\boldsymbol{E}$. Then switching to the frame moving to the $x$ direction with velocity $\boldsymbol{v}$, we have $\boldsymbol{E}' = \gamma(\boldsymbol{E} + \boldsymbol{v} \times \boldsymbol{B})$ and $\boldsymbol{B}' = \gamma(\boldsymbol{B} - \boldsymbol{v} \times \boldsymbol{E})$, where $\gamma$ is the usual gamma factor, $\gamma = \sqrt{1 - v^2}$. As a consequence, when $\boldsymbol{E} + \boldsymbol{v} \times \boldsymbol{B} = 0$, electric charges do not feel the electric field and thus are no longer accelerated. This situation is achieved when $\boldsymbol{v}$ is parallel to the direction of $\boldsymbol{E} \times \boldsymbol{B}$, and the magnitude is with $|\boldsymbol{v}| = v_x = E_y/B_z$. The electric current in this equilibrium situation (in the original frame) is $j_x = \sum_i \left(\frac{Q_i n_i}{B_z}\right) E_y$ with $Q_i$ and $n_i$ as the electric charge and number density of carrier particle $i$, respectively. In the weak field limit, $\sum_i Q_i n_i = \frac{e n_F}{6} \propto e p_F^3$ while in the high field limit $\sum_i Q_i n_i = \frac{e n_F}{3} \propto (eB) p_F$. We anticipate that the $1/B$ scaling of the Hall conductivity in the weak magnetic field comes from dissipative conductivity as $\sigma_{xy}^{\text{I, Fermi}} \to \frac{e n_F}{6B}$. Although we are not still able to

find a general expression for $a_1(M, q, \mu)$, in the limit of $M \to 0$, we could find the formula $a_1(0, q, \mu) = \frac{7e^3}{27\pi\mu}$, using the quark propagator expanded in power of magnetic field [34].

## 4. Concluding Remarks

We have discussed the transport properties of dense QCD matter in the magnetic field, based on the Kubo formula. In particular, we have paid attention to the quantum Hall effect by fully taking into account the Landau quantization. If there developed a new phase in the core of compact stars (an interesting phenomenon) the anomalous Hall effect is to be activated, aside from the usual Hall effect. The phase of dual chiral density wave may appear in the moderate density region of quark matter, where the energy spectrum of quarks resembles the one of electrons in Weyl semimetal in condensed-matter physics; the effective Hamiltonian exhibits the same structure as each other. Therefore, one may expect that some transport properties of quark matter in compact stars can be explored in the terrestrial experiments.

We have also paid some attention to the matter contribution to the quantum Hall conductivity. We derived an analytic formula for the non-dissipative part of the Hall conductivity. Based on this formula, we have examined its $B$ dependence in the limit, $q \to 0$. We have seen that the expected standard behavior of the Hall conductivity for conducting media, $\sigma_{xy}^{\text{Fermi}} \propto en/B$, that comes from non-dissipative part in the high field limit; when the magnetic field is very strong, the $B$ dependence of the Hall conductivity mostly comes from the LLL contribution and other higher Landau-levels decoupling. On the other hand, it should come totally from dissipative part in the opposite limit, where almost all the Landau levels contribute to the conductivity. Thus, the classical Drude–Zener formula becomes meaningless in the high field limit. One may expect a similar situation even in the phase of a dual chiral density wave.

Theoretically, we have found some geometric or topological effects. In particular, we have seen that the energy spectrum exhibits asymmetry with respect to the null line, and such spectral asymmetry plays an important role through the $\eta$ invariant in the presence of the magnetic field. The $\eta$ invariant is the topological one and leads to the anomalous particle number, and the anomalous Hall conductivity is proportional to it by way of the Streda formula. Consequently, we have shown that the anomalous Hall conductivity is independent of the magnetic field to give the same form as the one in the absence of the magnetic field as it should be.

We have put a special emphasis on a similarity between the DCDW phase and WSM, but there is a subtle difference between them; the expression of the anomalous Hall conductivity is different between them. We have seen that this difference may be originated from axial anomaly, but further discussions are needed to clarify it by way of the bulk-boundary correspondence, for example.

The magnetic-field dependence of the thermal Hall conductivity is phenomenologically important to understand thermal evolution of compact stars: an anisotropy of the thermal transport parallel and perpendicular to the magnetic field becomes important there. We have seen that the anomalous Hall effect should be dominant over the usual Hall effect in the strong magnetic field. We have also shown that the quantum contribution $\sigma_{xy}^{\text{II}}$ becomes essential, compared to the classical analog $\sigma_{xy}^{\text{I}}$ in the high field limit. These results suggest careful discussions of thermal evolutions of compact stars as a future work.

The purpose of our studies is to figure out the local properties of matter in compact stars by way of microscopic physic and to present physical inputs for basic equations to understand compact-star phenomena. For a global description of the properties of matter inside compact stars, one further needs to take into account the effects of general relativity and magnetohydrodynamics (MHD) [38]. For example, the anomalous Hall effect may affect MHD through the modifications of the Maxwell equations.

Our framework is fully relativistic and should be applicable even for magnetars, where the strength of the magnetic field exceeds the one of relativistic magnetic field of $4.414 \times 10^{13}$ G [39]. Herein, we have qualitatively discussed some features of the transport

phenomena, but further studies are needed for astrophysical applications of our results. Preparation of numerical tables of conductivity as functions of density, temperature, and magnetic field is left for a future work.

**Funding:** The work of H.A. was supported by JSPS KAKENHI Grant Number JP19K03868.

**Institutional Review Board Statement:** Not applicable.

**Informed Consent Statement:** Not applicable.

**Data Availability Statement:** The data presented in this study are available on request from the corresponding author.

**Acknowledgments:** One of the authors (T.T.) thanks the organizers for their hospitality during the conference "The Modern Physics of Compact Stars and Relativistic Gravity 2019". The work of H.A. was supported by JSPS KAKENHI Grant Number JP19K03868.

**Conflicts of Interest:** The authors declare no conflict of interest.

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
