# Peer review of "Transport Properties in Magnetized Compact Stars"

_2571-712X, doi:10.3390/particles4010009_

Round 1

Reviewer 1 Report

Referee report on "Transport properties in magnetized compact stars", by Toshitaka Tatsumi, Hiroaki Abuki, submitted to Particles

The present paper discusses the transport properties of the dense quark matter in the strong magnetic field on the bases of the Nambu-Jona-Lasinio (NJL) model concentrating on the anomalous Hall effect. The authors suppose that in the compact stars the anomalous Hall effect occurs in addition to the usual Hall effect and the dual chiral density wave phase may appear in the moderate density region. The paper presents some interest. I would suggest this manuscript to be published. Nevertheless, below there are a few comments or questions that I would suggest the authors to consider.

1.) The main subject of the paper is the quark matter. However, in the present work the problems of the quark matter are discussed very poorly and superficially. The paper is abounded by the condensed matter physics terms and deficiently reflects the quark-gluon plasma by itself.

2.) The authors should define all the quantities related to the NJL model more accurately and give the main points of formulation of this model. The concrete relativistic quantum fields should be discussed.

3.) For eqs. (1), (2) and (11), and (20) the proper references should be given.

Author Response

Thank our referee for careful reading of our manuscript.
We have revised the manuscript by taking into your comments.

1) We have discussed transport properties of quark matter by using the Nambu-Jona-Lasinio model for spontaneous breaking of chiral symmetry. However, one of the unique points suggested in the paper is the resemblance of physics here to Weyl semimetals 
in condensed matter physics. We believe it is interesting to see 
the same physics in the completely different fields. 
So, keeping in mind both systems throughout the manuscript, we add some comments about quark matter in relevant places, e.g. in lines 40, 41 or 59, 60.

2)This comment may be somewhat overlapped with the first one. 
We add some details of the NJL model in the relevant places, eg. in Eqs. (2), (11), (20), taking into account the referee's comment.

3)Some part of this comment may be also overlapped with the previous two comments. 
We improve the derivations and notations in our framework and add a new reference 33 in Eq. (20).

We hope that the revised manuscript is now suitable for publication in "Particles.

Reviewer 2 Report

\documentstyle[12pt]{article}
\textwidth 160 mm
\textheight 225 mm
\topmargin -25mm
\hoffset -1cm
\pagestyle{empty}
%\usepackage{color}

\begin{document}

\begin{center}
{\Large\bf {Referee report \\to the paper particles-1088226 entitled\\
{\it "\textit{Transport properties in magnetized compact stars}" \\
by Toshitaka Tatsumi and Hiroaki Abuki}}}
\end{center}

The authors of the paper particles-1088226 have studied transport properties of dense quark matter in the strong magnetic field,
Magnetic field dependence as well as density dependence of the Hall conductivity is discussed in the inhomogeneous chiral phase.
The authors claim that anomalous Hall effect is intrinsic to the inhomogeneous chiral phase and resembles the one in Weyl semimetals in condensed matter physics.
Some theoretical aspects inherent in anomalous Hall effect are also revealed.

i) The abstract of the paper has not been well written.
It would be better if the author could slightly revise the text of abstract in a form presenting the main results from the
main part of the paper.

ii) The conclusion is very short. I advise the authors to summarize in detail the obtained results with discussions.

iii) The quality of figures 3 and 4 does not meet the standards of the Particles journal and has to be definitely improved.

iv) More detailed analysis of the results has to be performed. Since the Particles journal is oriented on publication of
the papers related to physics the authors have to explore
possible astrophysical applications of the obtained results with concrete tables and plots produced.

v) My main criticism is related to the total ignorance by the authors of the effects of the strong gravitational and electromagnetic
fields. The calculations performed by the authors are standard ones and correct only in the Newtonian framework excluding
strong magnetic fields. The critical magnetic field is calculated as $4.414 \times 10^{13}$ Gauss and can not be
ignored when the magnetic field exceeds this value in the central part of the magnetized neutron stars.
In addition all calculations made in the paper for the transport phenmena have to be performed in framework of general relativity.

I do not think that the performed mathematical calculations in the Newtonian framework
for the transport properties would warrant a
publication in a high impact journal like the \textbf{Particles} specialized in publishing results
having physical and astrophysical importance. At the moment
it really contains unrelated mathematical results rather than
physical ones since the spacetime curvature and effects of the extra strong magnetic field have been ignored and
there are no real astrophysical applications.

Since the study of this manuscript is
indeed of interest in addressing the behaviour of the transport phenomena inside neutron stars I would recommend to the authors
to essentially revise it along my critical comments and resubmit the revised version to journal. Then I would consider it for publication in the \textbf{Particles} journal.
I hope the revised paper would be in more suitable form for publication in the \textbf{Particles} journal.

\end{document}

Author Response

Thank our referee for careful reading of our manuscript.
We have revised the manuscript by taking into your comments.
Our correspondence to the referee's comments is as follows in order of raised points.

 i) We revise the abstract, following the suggestion.

ii) Following the referee's suggestions, we revise the concluding remarks to make them as clearly as possible: we  rearrange, rewrite, remove and add some sentences with adding new references 38,39.

iii)We improve Figs,3,4 as clearly as possible.

iv) Our aim of the present paper is to figure out some features of the transport properties in quark matter, together with novel remarks regarding its intimate relation to Weyl semimetals in condensed matter physics. 
We have discussed local microscopic properties of quark matter and further studies about bulk systems, are of course, needed for astrophysical applications.
Our results, together with equation of state, should provide microscopic inputs for the basic equations such as magnetohydrodynamics to reveal the global properties of matter inside compact stars. 
Numerical tables of conductivity as the function of density and magnetic field also become necessary to this purpose, but this remains as a future work. 
We explain these points in the concluding remarks.

 v) This comment has some overlap with the previous point iv) as is mentioned above, our purpose is to figure out the local properties of quark matter of small volume, which scale is much less than the bulk scale of stars, but large enough in the scale of strong interaction, where we can safely discard the gravitational effect or global configuration of the magnetic field inside stars. They come in when we solve the basic equations to get bulk properties of stars.

Besides, our framework is fully relativistic and it is, we believe, applicable even for large magnetic field beyond the critical one. 
We add some explanation about these points in the concluding remarks.

We hope that the revised manuscript is now suitable for publication in "Particles”.

Round 2

Reviewer 2 Report

Referee report
to the revised version to paper particles-1088226 entitled
"Transport properties in magnetized compact stars"
by Toshitaka Tatsumi and Hiroaki Abuki

The authors of the revised version of paper particles-1088226  have made the requested
modifications along the comments provided in my report. The study of this manuscript is
mathematically correct and indeed of interest in addressing the behaviour of the transport properties in relatovistic stars. I recommend the revised version of the submitted paper for publication
in the Particles journal since it is more or less suitable for publication in the Particles journal.